# Hydrogen Peroxide Alleviates Salt Stress Effects on Gas Exchange, Growth, and Production of Naturally Colored Cotton

**DOI:** 10.3390/plants13030390

**Published:** 2024-01-28

**Authors:** Jackson Silva Nóbrega, Valéria Ribeiro Gomes, Lauriane Almeida dos Anjos Soares, Geovani Soares de Lima, André Alisson Rodrigues da Silva, Hans Raj Gheyi, Rafaela Aparecida Frazão Torres, Fellype Jonathar Lemos da Silva, Toshik Iarley da Silva, Franciscleudo Bezerra da Costa, Maila Vieira Dantas, Riselane de Lucena Alcântara Bruno, Reginaldo Gomes Nobre, Francisco Vanies da Silva Sá

**Affiliations:** 1Academic Unit of Agricultural Engineering, Federal University of Campina Grande, Campina Grande 58430-380, PB, Brazil; jacksonnobrega@hotmail.com (J.S.N.); andre.alisson@estudante.ufcg.edu.br (A.A.R.d.S.); hans.gheyi@ufcg.edu.br (H.R.G.); fellype.jonathar@estudante.ufcg.edu.br (F.J.L.d.S.); maila.vieira02@gmail.com (M.V.D.); 2Center for Agricultural Sciences, Federal University of Paraíba, Areia 58397-000, PB, Brazil; valeriaribeiro1996@gmail.com (V.R.G.); lane@cca.ufpb.br (R.d.L.A.B.); 3Academic Unit of Agrarian Sciences, Federal University of Campina Grande, Pombal 58840-000, PB, Brazilfranciscleudo@gmail.com (F.B.d.C.); 4Caraúbas Multidisciplinary Center, Universidade Federal Rural do Semi-Árido, Caraúbas 59780-000, RN, Brazil; reginaldo.nobre@ufersa.edu.br; 5Agricultural Sciences Center, State University of Paraíba, Catolé do Rocha 58884-000, PB, Brazil

**Keywords:** *Gossypium hirsutum* L., reactive oxygen species, salinity, semi-arid region

## Abstract

Cotton is one of the most exploited crops in the world, being one of the most important for the Brazilian Northeast. In this region, the use of irrigation is often necessary to meet the water demand of the crop. Water is often used from underground wells that have a large amount of salt in their constitution, which can compromise the development of crops, so it is vital to adopt strategies that reduce salt stress effects on plants, such as the foliar application of hydrogen peroxide. Thus, the objective of this study was to evaluate the effects of foliar application of hydrogen peroxide on the gas exchange, growth, and production of naturally colored cotton under salt stress in the semi-arid region of Paraíba, Brazil. The experiment was carried out in a randomized block design in a 5 × 5 factorial scheme, with five salinity levels of irrigation water—ECw (0.3, 2.0, 3.7, 5.4 and 7.1 dS m^−1^)—and five concentrations of hydrogen peroxide—H_2_O_2_ (0, 25, 50, 75 and 100 μM), and with three replicates. The naturally colored cotton ‘BRS Jade’ had its gas exchange, growth, biomass production, and production reduced due to the effects of salt stress, but the plants were able to produce up to the ECw of 3.97 dS m^−1^. Foliar application of hydrogen peroxide at the estimated concentrations of 56.25 and 37.5 μM reduced the effects of salt stress on the stomatal conductance and CO_2_ assimilation rate of cotton plants under the estimated ECw levels of 0.73 and 1.58 dS m^−1^, respectively. In turn, the concentration of 12.5 μM increased water-use efficiency in plants subjected to salinity of 2.43 dS m^−1^. Absolute and relative growth rates in leaf area increased with foliar application of 100 μM of hydrogen peroxide under ECw of 0.73 and 0.3 dS m^−1^, respectively. Under conditions of low water salinity (0.3 dS m^−1^), foliar application of hydrogen peroxide stimulated the biomass formation and production components of cotton.

## 1. Introduction

Cotton (*Gossypium hirsutum* L.) is one of the most important crops worldwide, being the main source of natural fiber and responsible for 35% of all fiber destined for and used in textile production in the world [1]. Brazil is the fourth largest producer of cotton, behind India, China, and the United States, with a production of 3150.1 thousand tons of lint cotton in the 22/23 season, with the Midwest and Northeast regions standing out as the largest producers with 74.9 and 22% of the national production, respectively [2].

The semi-arid region of Northeast Brazil has had irregular rainfall over the last few years, and when associated with high evapotranspiration rates, this results in water deficits for plants, which makes the use of irrigation indispensable for the success of agricultural production. Thus, many producers end up using water from underground reservoirs, especially in periods of drought, and this type of water may contain high levels of salts [3].

High levels of salts in irrigation water are a limiting factor for agricultural production and can trigger a series of negative effects on crop development. An excess of salts can induce a reduction in the osmotic potential of the soil, limiting the plant’s ability to absorb water and nutrients; it also causes ionic toxicity and nutritional imbalance and can trigger oxidative stress from the excessive production and accumulation of reactive oxygen species—ROS [4,5,6].

Salt stress can also alter physiological processes, compromising gas exchange and the production of photoassimilates [7,8,9]. In addition, it can affect the processes of cell division and expansion, consequently inhibiting the growth and therefore the yield of crops [10].

In cotton, high salinity can reduce gas exchange as well as plant growth, resulting in losses in fiber production and quality. Thus, it is necessary to identify cotton genotypes that achieve a better response to saline irrigation and salt-affected soils, in order to enhance exploitation in areas under stress conditions [11].

In this context, there is a growing search for strategies that reduce the deleterious effects of salt stress on plants, and foliar application of hydrogen peroxide (H_2_O_2_) stands out among them. H_2_O_2_ is a non-radical ROS that, at low concentrations, can act in the signaling and mediation of physiological processes, besides being able to induce plant tolerance to stress, such as salt stress [12]. In addition, H_2_O_2_ can increase the activity of antioxidant enzymes, as well as the production of organic compounds and proteins that support the antioxidant defense system of plants [13].

Thus, pre-treatment or exposure, such as foliar application with H_2_O_2_ in plants, can induce or modulate cross-tolerance, through the physiological processes involved, such as in the photosynthetic process, as well as in various pathways responsive to stress conditions, such as in ROS and antioxidant metabolism, reducing oxidative stress and lipid peroxidation [14].

Endogenous application of H_2_O_2_ is associated with overexpression of antioxidative enzymes and increased levels of ascorbate and gutathione, which are associated with the accumulation of osmoprotective substances with proline and soluble carbohydrates, improving plant tolerance to stress conditions [15].

The beneficial effect of exogenous application of H_2_O_2_ as an attenuator of salt stress has been reported in the literature for several species, such as wheat [16], melon [17], tomato [18], maize [19], cucumber [20], rice [21], passion fruit [22], and bell pepper [23].

The hypothesis of the present study is that foliar application of H_2_O_2_ is able to reduce the deleterious effects of salt stress on naturally colored cotton through the regulation of physiological processes, contributing to a greater activity of the antioxidant defense system, improving photosynthetic activity and stomatal regulation, as well as reducing the oxidative stress triggered by the accumulation of ROS, promoting improvements in cotton growth and yield. Despite being a culture that has great importance for semi-arid regions, such as the Brazilian Northeast, little information is available on strategies such as foliar application of H_2_O_2_ to improve the performance of fiber cotton plants under naturally occurring saline stress conditions, which motivated the development of this research. Thus, the objective was to evaluate the effects of foliar application of H_2_O_2_ on the mitigation of salt stress effects on the gas exchange, growth, and production of the naturally colored cotton ‘BRS Jade’ in the semi-arid region of Paraíba, Brazil.

## 2. Results

There was a significant effect from the interaction between irrigation water salinity and H_2_O_2_ concentrations (*p* ≤ 0.05) on the stomatal conductance (*gs*) (*p* ≤ 0.01), transpiration (*E*), internal CO_2_ concentration (*Ci*), CO_2_ assimilation rate (*A*), water-use efficiency (*WUE*), and intrinsic carboxylation efficiency (*CEi*) of cotton plants at 35 days after sowing (DAS) (Table 1).

For stomatal conductance (*gs*), foliar application of H_2_O_2_ promoted improvements in the stomatal opening of cotton plants, with the highest value of *gs* (0.2067 mol H_2_O m^−2^ s^−1^) in plants subjected to a concentration of 56.25 µM and low ECw of 0.73 dS m^−1^ (Figure 1A). On the other hand, the lowest *gs* (0.1301 mol H_2_O m^−2^ s^−1^) was observed in plants subjected to ECw of 7.1 dS m^−1^ and without H_2_O_2_ application (0 µM), with a reduction of 37% when comparing the highest and lowest values obtained.

Cotton transpiration (*E*) was also favored by the foliar application of H_2_O_2_, with the highest value of 4.97 mmol H_2_O m^−2^ s^−1^ under to ECw of 0.3 dS m^−1^ and at a concentration of 50 μM (Figure 1B). Thus, as observed for *gs*, the increase in salinity caused a reduction in *E*, with the lowest value (3.51 mmol H_2_O m^−2^ s^−1^) obtained in the control treatment (0 μM H_2_O_2_) and under ECw of 7.1 dS m^−1^, resulting in a decrease of 29.4% when comparing the maximum and minimum values obtained.

For the internal CO_2_ concentration (*Ci*), the highest value (203.61 μmol CO_2_ m^−2^ s^−1^) occurred in plants subjected to the lowest level of water salinity (0.3 dS m^−1^) and the H_2_O_2_ concentration of 62.5 μM (Figure 1C), followed by decreases with the increase in ECw, and the lowest value (176.71 μmol CO_2_ m^−2^ s^−1^) was obtained in plants that did not receive H_2_O_2_ application (0 μM) and were irrigated with 4.5 dS m^−1^ water.

Foliar application of H_2_O_2_ reduced the deleterious effect of salt stress on the CO_2_ assimilation rate (*A*) of cotton, with the highest value (18.95 μmol CO_2_ m^−2^ s^−1^) obtained under an estimated ECw of 1.58 dS m^−1^ and H_2_O_2_ concentration of 37.5 μM (Figure 1D). The increase in ECw up to 7.1 dS m^−1^ and H_2_O_2_ up to the concentration of 100 μM caused the lowest value of *A* (9.26 μmol CO_2_ m^−2^ s^−1^), leading to a reduction of 51.1% compared to the maximum estimated values.

Water-use efficiency (*WUE*) increased with the application of H_2_O_2_ up to a concentration of 12.5 μM and under the estimated water salinity of 2.43 dS m^−1^, with a value of 3.97 [(μmol CO_2_ m^−2^ s^−1^) (mol H_2_O m^−2^ s^−1^)^−1^] (Figure 1E). From this point on, there was a downward trend in *WUE*, with the lowest value (2.66 [(μmol CO_2_ m^−2^ s^−1^) (mol H_2_O m^−2^ s^−1^)^−1^] in plants subjected to ECw of 7.1 dS m^−1^ and 100 μM of H_2_O_2_, representing a reduction of 33% when compared to the highest value of *WUE*.

The highest intrinsic carboxylation efficiency (*CEi*) occurred in plants cultivated under an estimated ECw of 1.15 dS and 0 μM of H_2_O_2_, being 0.1030 [(μmol CO_2_ m^−2^ s^−1^) (μmol CO_2_ m^−2^ s^−1^)^−1^] (Figure 1F). The increase up to the maximum concentration of H_2_O_2_ and salinity led to the lowest *CEi* (0.0501 (μmol CO_2_ m^−2^ s^−1^) (μmol CO_2_ m^−2^ s^−l^)^−1^), resulting in a reduction of 51.3% when the maximum and minimum values were compared.

There was a significant effect (*p* ≤ 0.01) from the interaction between irrigation water salinity and H_2_O_2_ concentrations (Table 2) on the absolute growth rates in plant height (AGR_PH_) and leaf area (AGR_LA_) and relative growth rate in the leaf area (RGR_LA_) of cotton plants (*p* ≤ 0.05). Water salinity had individual effects (*p* ≤ 0.01) on the absolute growth rate in stem diameter (AGR_SD_) and relative growth rate in plant height (RGR_PH_). Foliar application of H_2_O_2_ significantly affected the absolute growth rate in stem diameter (AGR_SD_) in the period of 35–110 days after sowing.

For AGR_PH_, foliar application of H_2_O_2_ up to a concentration of 100 μM increased the growth in height of cotton plants under low salinity (0.3 dS m^−1^), which led to the highest value (0.6723 cm day^−1^) (Figure 2A). The increase in salinity reduced AGR_PH_, and the lowest estimated value of 0.2685 cm day^−1^ was obtained in plants cultivated under ECw of 7.1 dS m^−1^ and 25 μM of H_2_O_2_, representing a reduction of 60% compared to the highest value obtained.

The increase in salinity levels also reduced RGR_PH_, with the lowest value (0.0079 cm cm^−1^ day^−1^) observed in plants subjected to ECw of 6.0 dS m^−1^, with a decrease of 55.2% compared to those submitted to ECw of 0.3 dS m^−1^ (0.017 cm cm^−1^ day^−1^) (Figure 2B).

For AGR_SD_, water salinity caused a decreasing linear effect, with reduction of 60.9% when plants under the lowest ECw level (0.3 dS m^−1^) were compared with those subjected to the highest water salinity (Figure 2C). In turn, the H_2_O_2_ concentrations caused a quadratic effect, with the highest estimated value (0.07852 mm day^−1^) with foliar application of 13 mM H_2_O_2_, followed by decreases with the increase in concentration (Figure 2D).

Regarding the absolute and relative growth rates in leaf area (AGR_LA_ and RGR_LA_), H_2_O_2_ reduced the deleterious effect of salt stress, with the highest values (16.94 cm² day^−1^ and 0.0262 cm cm^−2^ day^−1^) obtained in plants subjected to estimated ECw of 0.73 and 0.3 dS m^−1^ and H_2_O_2_ concentrations of 100 μM, respectively (Figure 2E,F). On the other hand, the lowest values of AGR_LA_ and RGR_LA_ (3.53 cm² day^−1^ and 0.0125 cm cm^−2^ day^−1^) were observed in plants under ECw of 7.1 dS m^−1^ and H_2_O_2_ concentrations of 25 and 37.5 μM, representing decreases of 79.1 and 52.3% in AGR_LA_ and RGR_LA_, respectively.

There was a significant effect (*p* ≤ 0.01) from the interaction between water salinity levels and H_2_O_2_ concentrations for root, stem, leaf, and total dry mass of cotton plants (Table 3) at 130 days after sowing.

For root dry mass (Figure 3A), the highest value (11.03 g per plant) was observed in plants under ECw of 0.3 dS m^−1^ and H_2_O_2_ concentration of 100 μM. On the other hand, the lowest value (2.65 g per plant) was reached in plants cultivated under salinity of 7.1 dS m^−1^ and H_2_O_2_ concentration of 31.25 μM, with a decrease of 76% when comparing the maximum and minimum values obtained.

As observed for root dry mass, the maximum value of SDM (24.95 g per plant) occurred in plants submitted to ECw of 0.3 dS m^−1^ and H_2_O_2_ concentration of 100 μM (Figure 3B). Irrigation with ECw of 7.1 dS m^−1^ and H_2_O_2_ concentration of 50 μM led to the lowest value (6.20 g per plant), which represents a reduction of 75.2% when compared to the maximum value obtained.

For leaf dry mass (Figure 3C), the highest estimated value (12.21 g per plant) was achieved in plants subjected to ECw of 0.3 dS m^−1^ and H_2_O_2_ concentration of 56.25 μM. The increase in salinity promoted a marked reduction in LDM, with the minimum value (3.05 g per plant) obtained at ECw of 7.1 dS m^−1^ and H_2_O_2_ concentration of 0 μM.

For the total dry mass of cotton plants (Figure 3D), the application of 100 μM of H_2_O_2_ promoted the highest accumulation (47.24 g per plant) under water salinity of 0.3 dS m^−1^. Foliar application of H_2_O_2_ at concentration of 31.25 μM and water salinity of 7.1 dS m^−1^ resulted in the lowest TDM accumulation (12.65 g per plant), with a decrease of 73.2% when the maximum and minimum values were compared.

The interaction between water salinity levels and H_2_O_2_ concentrations (Table 4) had a significant effect (*p* ≤ 0.01) on the production components of naturally colored fiber cotton.

The highest number of bolls per plant (16 bolls per plant) occurred in plants produced under ECw of 0.3 dS m^−1^ and 0 μM of H_2_O_2_ (Figure 4A). The increase in water salinity and H_2_O_2_ concentrations caused a severe reduction in NB, and the lowest value (1 boll) was observed in plants subjected to ECw of 7.1 dS m^−1^ and foliar application of 100 μM of H_2_O_2_, representing a reduction of 93.3% when compared to the highest value obtained, indicating that the production was compromised by water salinity and H_2_O_2_.

For the average boll weight (Figure 4B), the highest value (2.55 g per boll) was obtained in plants subjected to ECw of 0.3 dS m^−1^ and H_2_O_2_ concentration of 50 μM. On the other hand, the lowest value obtained (1.41 g per boll) was observed in plants subjected to 5.4 dS m^−1^ and 0 μM of H_2_O_2_, representing a reduction of 44.7% compared to the maximum estimated value.

Seed cotton weight was higher (32.08 g per plant) in plants subjected to ECw of 0.3 dS m^−1^ and H_2_O_2_ concentration of 56.25 μM (Figure 4C). The increase in salinity up to 7.1 dS m^−1^ caused a severe reduction in SCW, with the lowest value (1.65 g per plant) being observed in plants that did not receive H_2_O_2_ (0 μM), resulting in a decrease of 94.8% when comparing the maximum and minimum values.

Regarding lint cotton weight (Figure 4D), the highest value (16.69 g per plant) was obtained under salinity of 0.3 dS m^−1^ and H_2_O_2_ concentration of 50 μM. As ECw and H_2_O_2_ were increased, lint cotton weight was reduced, and the lowest value (0.85 g per plant) was observed under ECw of 7.1 dS m^−1^ and H_2_O_2_ concentration of 100 μM, resulting in a reduction of 94.9% compared to the highest value obtained.

Fiber yield reached its highest value of 58.84% in plants cultivated under ECw of 0.3 dS m^−1^ and foliar application of 25 μM of H_2_O_2_ (Figure 4E), whereas the lowest yield (41%) was obtained in plants subjected to ECw of 3.7 dS m^−1^ and H_2_O_2_ concentration of 100 μM.

## 3. Discussion

The high salt content in irrigation water compromises the metabolism of cotton plants, causing reductions in gas exchange, growth, and production. This effect occurs because salt stress triggers changes in biochemical, physiological, and morphological processes [24]. Among these effects, the reduction in osmotic potential stands out, in its restriction of the ability of plants to absorb water and nutrients [25]. Excess salt ions can also cause ionic toxicity and nutritional imbalance, limiting the growth and physiological processes of plants [4,26].

In the present study, there was a marked reduction in gas exchange in plants subjected to water salinity of 7.1 dS m^−1^, reinforcing the deleterious effect of salt stress on the cotton. The stomatal limitation caused by salt stress affected the *gs*, reducing transpiration and internal CO_2_ concentration. This effect occurs because plants close their stomata as a defense mechanism against water loss, which reduces transpiration and CO_2_ assimilation [27,28]. Furthermore, this stomatal regulation mechanism leads to a reduction in the flow of water vapor, aiming to maintain water potential and prevent dehydration of guard cells, which ends up inducing a lower influx of CO_2_ into leaf mesophyll cells, reducing transpiration [29].

This negative effect caused by salt stress on gas exchange has also been reported for other species, as observed by Souza et al. [24] in zucchini, for which ECw above 1.0 dS m^−1^ drastically reduces gas exchange. Hamani et al. [30] reported that the gas exchange of cotton plants was reduced by 150 mM of NaCl. Rodrigues et al. [31] found that salinity above 1.1 dS m^−1^ causes reduction in gas exchange in sunflower plants.

It is worth pointing out that in plants subjected to foliar application of H_2_O_2_, there was an improvement in gas exchange, with increments in the *gs* and CO_2_ assimilation rate of the cotton plants. H_2_O_2_ is able to induce tolerance to stress through the signaling and modulation of the plant’s antioxidant defense system [6]. At low levels, hydrogen peroxide has the ability to quickly diffuse into the membrane, achieving the coordination of several signaling molecules, which increases the accumulation of osmoprotective substances and improves the plant’s antioxidative defense system [32,33], as observed in our research, where gas exchange was better at levels close to 50 µM.

Beneficial effects from the foliar application of H_2_O_2_ have been reported by other authors, such as Veloso et al. [34], who worked with colored cotton cv. ‘BRS Rubi’ and found that the concentration of 50 μM promoted an increase in gas exchange in plants irrigated with water of 5.3 dS m^−1^. In bell pepper, Aragão et al. [23] observed that the H_2_O_2_ concentration of 15 μM promoted increments in the stomatal conductance, CO_2_ assimilation rate, and carboxylation efficiency of plants subjected to salinity of up to 0.8 dS m^−1^.

H_2_O_2_ application improved water-use efficiency in plants subjected to an estimated ECw of 2.43 dS m^−1^, demonstrating its beneficial effect on cotton under saline conditions. A similar result was observed by Capitulino et al. [35] in *Annona muricata* L., as the concentration of 10 μM increased water-use efficiency in plants cultivated under ECw of 1.8 dS m^−1^. This effect is attributed to the fact that H_2_O_2_ acts on the signaling of the plant’s antioxidant defense system.

The deleterious effect caused by salt stress inhibited the absolute and relative growth rates, limiting the processes of cell division and expansion, which compromised the development of cotton plants. Thus, as it affects these cell expansion processes, salt stress affects the architecture of cotton plants, reducing growth rates as well as the accumulation of biomass in plant organs. This behavior is linked not only to the ionic and osmotic effect, but also to oxidative stress, triggered by the generation and excessive accumulation of reactive oxygen species [36]. This result has been observed by Tabassum et al. [10] in pea, Gohari et al. [37] in basil, and Aragão et al. [23] in bell pepper.

It is worth pointing out that in this study, the application of 100 μM of H_2_O_2_ increased the AGR_LA_ and RGR_LA_ of cotton plants, demonstrating its beneficial role in protecting against deleterious effects of salt stress, due to its action as a signaling molecule, inducing the regulation of metabolic processes that increase the tolerance of plants to salinity, as observed in pea (*Pisum sativum* L.) by Dito and Gadallah [13]. In addition, H_2_O_2_ can induce enzymatic activity, improve membrane stability, and act on the metabolism of lipids, fatty acids, and carbohydrates, increasing the plant’s tolerance to stress [12].

The biomass production of the plants was drastically reduced by salt stress, standing out as a result of the decrease in the production and accumulation of photoassimilates in the different organs. The excess of salts is a limiting factor for plant development, due to the osmotic, ionic phytotoxicity, and oxidative stress effects, which inhibit the processes of cell elongation and expansion [38].

In the present study, the application of H_2_O_2_ up to a concentration of 100 μM stimulated biomass production in plants subjected to ECw of 0.3 dS m^−1^, with a reduction as irrigation water salinity increased. This result has been observed in bell pepper [23] and zucchini [39] cultivated using saline nutrient solutions in a hydroponic system.

The effects of salt stress observed on gas exchange and growth affected the production components of cotton, with a marked reduction in the number of bolls, seed, and lint cotton weights and fiber yield. The reduction in cotton production triggered by salt stress is mainly due to the lower emergence of flower buds and fall of flowers and bolls, resulting from the osmotic and ionic effects of salinity [40]. According to these authors, salt stress, in addition to reducing production and lint yield of cotton, can also reduce fiber quality, affecting its length, strength, and maturity.

It is worth highlighting that, even under high salinity, ‘BRS Rubi’ naturally colored cotton plants have a certain tolerance to salt stress, being able to produce up to ECw of 3.97 dS m^−1^. Cotton tolerance to salt stress involves variable responses that are directly associated with the expression of genes that regulate the plant’s defense system. In addition, the response is influenced by factors such as genotype and developmental stage, as the accumulation of salts in the different physiological stages can have consequences for cotton yield, as observed by Shaheen et al. [41].

However, it must be highlighted that the application of H_2_O_2_ at concentrations of 25 and 50 μM increased the production components in plants subjected to water with an electrical conductivity of 0.3 dS m^−1^. Similarly, Veloso et al. [42] studied colored cotton cv. ‘BRS Jade’ grown under salt stress and observed that the application of 75 μM of H_2_O_2_ increased the seed cotton and lint cotton weights under irrigation with 0.8 dS m^−1^ water. According to these authors, under these conditions there was an improvement in fiber quality, with an increase in fiber strength, micronaire index, and maturity.

## 4. Materials and Methods

### 4.1. Study Location

The experiment was conducted in pots under open field conditions during the period from July to November 2019 at the Center for Sciences and Agri-Food Technology, CCTA of the Federal University of Campina Grande, UFCG, in Pombal, Paraíba, Brazil, located at the geographic coordinates 6°47′20″ South latitude and 37°48′01″ West longitude, at an altitude of 194 m. The climate of the region is hot and dry semi-arid, with an average annual evaporation of 2000 mm and an average rainfall of approximately 750 mm year^−1^, according to Köppen’s climate classification adapted to Brazil [43]. The local meteorological data collected during the experimental period are shown in Figure 5. 

### 4.2. Treatments and Statistical Design

The statistical design consisted of randomized blocks in a 5 × 5 factorial scheme, corresponding to five levels of salinity of irrigation water—ECw (0.3, 2.0, 3.7, 5.4, and 7.1 dS m^−1^) and five concentrations of hydrogen peroxide—H_2_O_2_ (0, 25, 50, 75 and 100 μM), with three replicates, totaling 75 plots. ECw levels were defined based on a study conducted by Lima et al. [44], whereas H_2_O_2_ concentrations were based on a study conducted by Silva et al. [45]. To ensure that the plants were under the indicated salinity conditions, the electrical conductivity of the soil was measured every 15 days through the drainage water with the aid of a bench conductivity meter.

### 4.3. Experiment Conduction

The naturally colored cotton genotype ‘BRS Jade’, considered a moderately salinity-tolerant cultivar, was sourced from the National Cotton Research Center (CNPA) of Embrapa Cotton for the study. The cultivar has light brown fiber and is adapted to Cerrado and semi-arid conditions, presenting productivity and good fiber characteristics, such as length, uniformity, resistance, and micronaire index. Furthermore, it is resistant to some of the main cotton diseases, such as angular leaf spot (*Xanthomonas citri* subsp. *malvacearum*), and moderately resistant to common mosaic virus (Abutilon mosaic virus—AbMV) [46].

For sowing, five seeds were planted in each pot at 3 cm depth, in plastic pots adapted as drainage lysimeters with capacity of 20 L. The lysimeters were filled with 0.5 kg of gravel, and the base of each was drilled and connected to a 15-millimetre-diameter drain, which was wrapped with a non-woven geotextile (Bidim OP 30) to prevent clogging by soil material. A plastic bottle was attached to each drain to collect drained water, which was used to determine water consumption by the plants. The pots were arranged in single rows spaced 1.2 m apart, with 1.0 m between plants in the row.

The soil used was a *Neossolo Regolítico Eutrófico* (Entisol) with sandy loam texture, from an agricultural area in the municipality of Pombal-PB. It was collected at 0–20 cm depth, the following physical–hydraulic and chemical attributes being determined according to the methodology of Texeira et al. [47]: pH (H_2_O) = 5.58; organic matter = 2.93 g kg^−1^; P = 39.2 mg kg ^−1^; K^+^ = 0.23 cmol_c_ kg^−1^; Na^+^ = cmol_c_ kg^−1^; Ca^2+^ = 9.07 cmol_c_ kg^−1^; Mg^2+^ = 2.78 cmol_c_ kg^−1^; Al^3+^ = 0.0 cmol_c_ kg^−1^; H^+^ = 8.61 cmol_c_ kg^−1^. EC_se_ = 2.15 dS m^−1^; CEC = 22.33 cmol_c_ kg^−1^; SAR_se_ = 0.67 mmol L^−1^; ESP = 7.34%; Particle-size fraction: Sand = 272.7 g kg^−1^; Silt = 100.7 g kg^−1^; Clay = 326.6 g kg^−1^. Moisture 32 dag kg^−1^ = 25.91 KPa and 1519.5 dag kg^−1^ = 12.96 KPa.

NPK fertilization was carried out according to the recommendation of Novais et al. [48] for pot experiments, using 100, 300, and 150 mg kg^−1^ of N, P_2_O_5,_ and K_2_O. Urea, monoammonium phosphate, and potassium chloride were used as sources of nitrogen, phosphorus, and potassium, respectively. Fertilization was carried out as topdressing via irrigation water at 18, 38, and 58 days after sowing (DAS) equitably. The source of micronutrients was Dripsol Micro^®^ [SQM VITAS, Candeias-BA, Brazil] (1.2% Mg, 0.85% B, 3.4% Fe, 4.2% Zn, 3.2% Mn, 0.5% Cu, 0.06% Mo), at a concentration of 1.5 g L^−1^, applied every 10 days after emergence and subsequently supplied at 15-day intervals.

### 4.4. Water Preparation and Irrigation Management

The irrigation waters were obtained from the addition of NaCl, CaCl_2_·2H_2_O, and MgCl_2_·6H_2_O salts, according to the pre-established treatments, in water from the local supply system (Pombal-PB). The amount of each compound was determined considering the relationship between ECw and the concentration of salts [49].

After preparing the waters, the ECw level was checked, and if necessary, adjusted before use.

Prior to sowing, the volume of water needed to raise the soil moisture content to the level corresponding to field capacity was determined, and water was applied according to the established treatments. Irrigation was carried out daily at 5 p.m., applying to each lysimeter the volume corresponding to that obtained by the water balance, and the volume of water to be applied was determined.

### 4.5. Preparation and Application of Hydrogen Peroxide Concentrations

The solutions were prepared by diluting H_2_O_2_ in deionized water at each application event and were maintained under dark conditions to avoid degradation in the presence of light. H_2_O_2_ applications began at 15 DAS and were later carried out weekly until the end of the crop cycle, with spraying conducted in such a way as to fully wet the leaves on the abaxial and adaxial sides, using a spray bottle, from 5 p.m. The plants were isolated with a plastic tarpaulin structure during the H_2_O_2_ applications to prevent the solutions from drifting, and an average volume of 15 mL per plant was applied in each application.

### 4.6. Variables Analyzed

Leaf gas exchange was evaluated at 35 DAS using a portable infrared gas analyzer—IRGA (Infrared Gas Analyser, model LCpro—SD, from ADC Bioscientific, UK, Hoddesdon, UK). Stomatal conductance—*gs* (mol CO_2_ m^−2^ s^−1^), transpiration—*E* (mmol H_2_O m^−2^ s^−1^), CO_2_ assimilation rate—*A* (μmol CO_2_ m^−2^ s^−1^), and internal CO_2_ concentration—*Ci* (μmol CO_2_ m^−2^ s^−1^) were determined. Water-use efficiency (*WUE*) was obtained through the A/E ratio [(µmol CO_2_ m^−2^ s^−1^) (mol H_2_O m^−2^ s^−1^)^−1^], and instantaneous carboxylation efficiency (*CEi*) from the A/Ci ratio [(µmol CO_2_ m^−2^ s^−1^) (μmol CO_2_ m^−1^ s^−1^)^−1^]. The determinations were carried out between 6:30 a.m. and 9:30 a.m. on the third fully expanded and photosynthetically active leaf, counted from the apical bud, under natural conditions of air temperature and CO_2_ concentration and using an artificial radiation source of 1200 μmol m^−2^ s^−1^, established through the light–photosynthesis response curve.

Growth was evaluated by plant height, measured with a ruler graduated in cm, stem diameter was measured with a digital caliper, and leaf area was obtained by measuring leaf width and length at 35 and 110 DAS. From these data, the absolute (AGR) and relative (RGR) growth rates in plant height, stem diameter, and leaf area were determined using the methodology of Benincasa [50].

At 130 DAS, the plants were removed from the lysimeters, and the dry mass measurements of leaves, stems, and roots were determined using the drying method in an oven with air circulation at 65 °C, where the parts were packed in kraft paper bags and dried until they reached constant weight. Total dry mass of cotton plants was obtained from the sum of all biomass components.

In the same period, the production components were quantified based on: number of bolls (NB), obtained by counting the number of bolls harvested per plant; seed cotton weight (SCW), obtained by weighing all the cotton harvested; average boll weight (ABW), obtained from the ratio between SCW and NB; lint cotton weight (LCW), obtained by weighing the fibers after removing the seeds; and fiber yield, calculated from the ratio of LCW to SCW.

The collected data were subjected to the normality test (Shapiro–Wilk), followed by analysis of variance using the F test at *p* ≤ 0.05%. In cases where there was a significant effect, polynomial regression analysis (*p* ≤ 0.05) was applied to ECw levels and H_2_O_2_ concentrations, using the statistical program Sisvar—ESAL version 5.6 [51]. The results were presented using response surface, and Sigma Plot ^®^ 12.5 software was used to construct the graphs.

## 5. Conclusions

The naturally colored cotton ‘BRS Jade’ had its gas exchange, growth rates, biomass production, and production components compromised by the increase in the salinity of irrigation water. However, the plants were able to produce up to an ECw of 3.97 dS m^−1^. It is worth pointing out that the application of H_2_O_2_ attenuates the effect of salinity under stomatal conductance, CO_2_ assimilation rate, and water-use efficiency of ‘BRS Jade’ cotton. On the other hand, foliar application of 100 μM of H_2_O_2_ stimulates the gas exchange, growth, and production of cotton under irrigation with 0.3 dS m^−1^ water. Thus, the results obtained reinforce the idea that H_2_O_2_ promotes improvements in physiological processes through its action in signaling the antioxidant defense system of the plant, proving it to be a viable alternative to increase plant tolerance to the effects of salt stress, especially in regions with water limitation, such as the Brazilian semi-arid region. However, it is important to highlight that further studies should be conducted to better understand the mode of action of H_2_O_2_ in the signaling of the defense system, so research focused on biochemical and enzymatic activity is needed.

## Figures and Tables

**Figure 1 plants-13-00390-f001:**
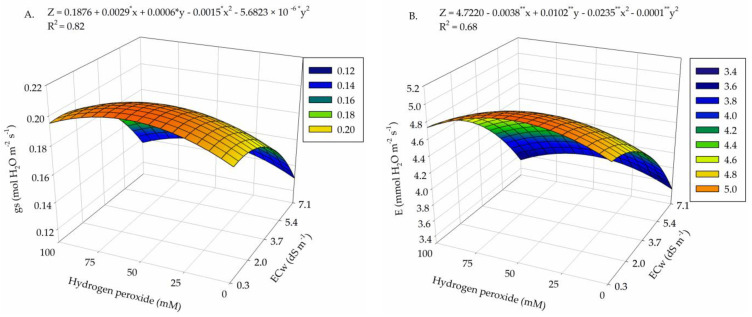
Stomatal conductance—*gs* (**A**), transpiration—*E* (**B**), internal CO_2_ concentration*—Ci* (**C**), CO_2_ assimilation rate—*A* (**D**), water-use efficiency—WUE (**E**), and intrinsic carboxylation efficiency*—CEi* (**F**) of cotton as a function of the interaction between irrigation water salinity levels—ECw and hydrogen peroxide concentrations—H_2_O_2_ at 35 days after sowing. X and Y—Electrical conductivity of water—ECw and concentration of hydrogen peroxide—H_2_O_2_, respectively. *, **—significant at *p* ≤ 0.05, and significant at *p* ≤ 0.01 by F-test, respectively.

**Figure 2 plants-13-00390-f002:**
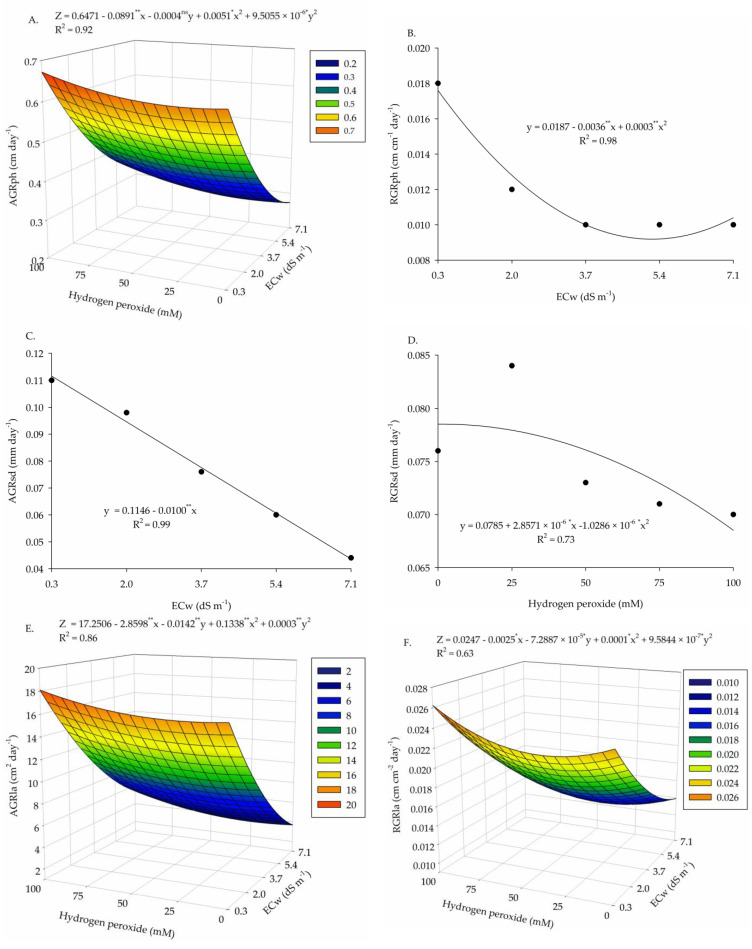
Absolute and relative growth rates of plant height—AGR_PH_ and RGR_PH_ (**A**,**B**), stem diameter—AGR_SD_ and RGR_SD_ (**C**,**D**), and leaf area—AGR_LA_ and RGR_LA_ (**E**,**F**) of cotton plants as a function of the interaction between salinity levels of irrigation water—ECw and hydrogen peroxide concentrations—H_2_O_2_, in the period 35–110 days after sowing. X and Y—Electrical conductivity of water—ECw and concentration of hydrogen peroxide—H_2_O_2_, respectively. ns, *, **—not significant, significant at *p* ≤ 0.05, and significant at *p* ≤ 0.01 by F-test, respectively.

**Figure 3 plants-13-00390-f003:**
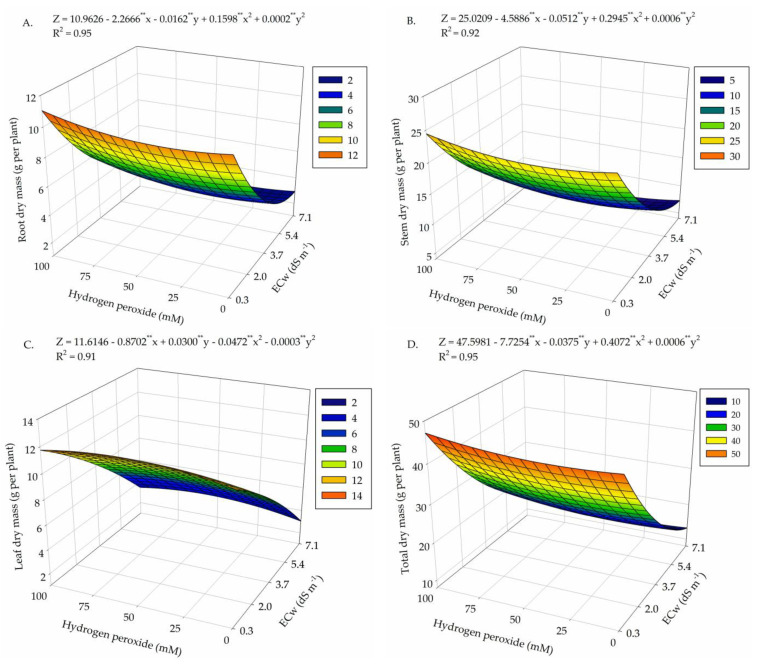
Root dry mass (**A**), stem dry mass (**B**), leaf dry mass (**C**), and total dry mass (**D**) of cotton plants as a function of the interaction between the salinity levels of irrigation water (ECw) and concentrations of hydrogen peroxide (H_2_O_2_) at 130 days after sowing. X and Y—Electrical conductivity of water—ECw and concentration of hydrogen peroxide—H_2_O_2_, respectively. **—significant at *p* ≤ 0.01 by F-test, respectively.

**Figure 4 plants-13-00390-f004:**
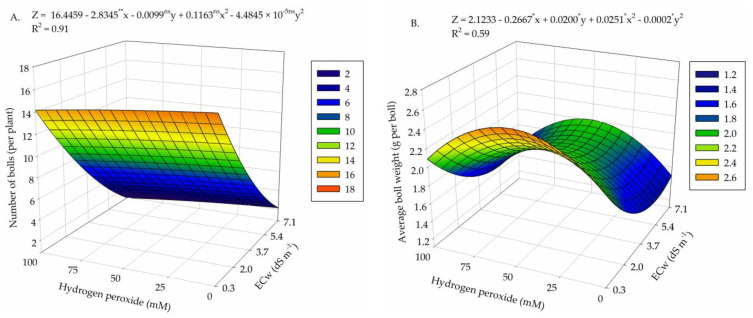
Number of bolls (**A**), average boll weight (**B**), seed cotton weight (**C**), lint cotton weight (**D**), and fiber yield (**E**) of cotton as a function of the interaction between irrigation water salinity levels (ECw) and hydrogen peroxide (H_2_O_2_) concentrations at 130 days after sowing. X and Y—electrical conductivity of water—ECw, and concentration of hydrogen peroxide—H_2_O_2_, respectively. ns, *, **—not significant, significant at *p* ≤ 0.05, and significant at *p* ≤ 0.01 by F-test, respectively.

**Figure 5 plants-13-00390-f005:**
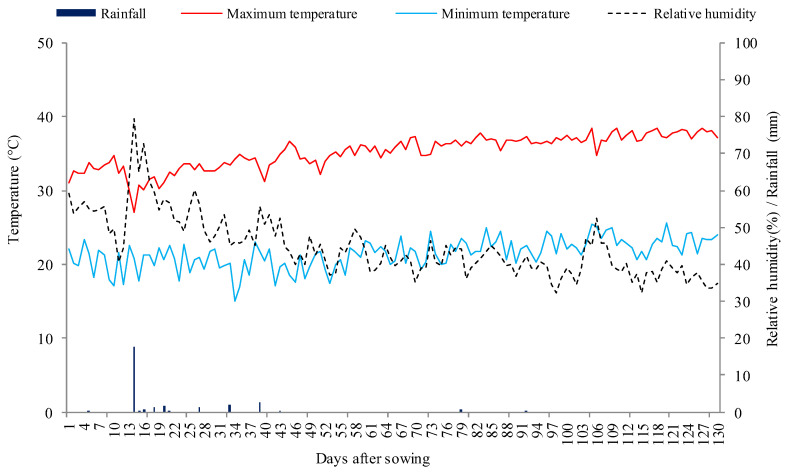
Meteorological data collected while conducting the experiment.

**Table 1 plants-13-00390-t001:** Summary of the analysis of variance for stomatal conductance (*gs*), transpiration (*E*), internal CO_2_ concentration (*Ci*), CO_2_ assimilation rate (*A*), water-use efficiency (WUE), and intrinsic carboxylation efficiency (*CEi*) of cotton grown under irrigation water salinity and hydrogen peroxide concentrations, at 35 days after sowing.

Variation Sources	DF	Mean Squares					
		*gs*	*E*	*Ci*	*A*	*WUE*	*CEi*
Salinity levels (SL)	4	9.1 × 10^−3^ **	5.49 **	348.55 ^ns^	158.33 **	2.21 **	4.4 × 10^−3^ **
Linear regression	1	3.2 × 10^−2^ **	15.15 **	305.30 ^ns^	525.54 **	5.76 **	1.4 × 10^−2^ **
Quadratic regression	1	3.5 × 10^−3^ **	0.98 **	12.87 ^ns^	76.50 **	2.30 *	2.0 × 10^−3^ **
Hydrogen peroxide (H_2_O_2_)	4	1.3 × 10^−3^ *	0.46 **	1124.78 ^ns^	39.73 **	1.05 ^ns^	1.4 × 10^−3^ *
Linear regression	1	1.0 × 10^−4 ns^	0.031 ^ns^	1485.22 ^ns^	22.41 ^ns^	0.96 ^ns^	1.3 × 10^−3 ns^
Quadratic regression	1	3.6 × 10^−3^ **	1.077 **	2474.28 ^ns^	39.34 *	0.43 ^ns^	8.0 × 10^−5 ns^
Interaction (SL × H_2_O_2_)	16	1.0 × 10^−3^ *	0.65 **	1923.74 **	26.51 **	1.15 **	1.2 × 10^−3^ **
Blocks	2	6.0 × 10^−4 ns^	0.18 ^ns^	88.04 ^ns^	0.288 ^ns^	0.085 ^ns^	5.0 × 10^−6 ns^
Residual	74	4.0 × 10^−4^	0.086	648.02	8.89	0.469	4.2 × 10^−4^
CV (%)		12.0	6.73	13.2	15.5	19.3	24.7

** significant at *p* ≤ 0.01; * significant at *p* ≤ 0.05; ^ns^ not significant.

**Table 2 plants-13-00390-t002:** Summary of the analysis of variance for the absolute and relative growth rates in plant height (AGR_PH_ and RGR_PH_), stem diameter (AGR_SD_ and RGR_SD_), and leaf area (AGR_LA_ and RGR_LA_) of cotton subjected to irrigation water salinity and hydrogen peroxide concentrations in the period 35–110 days after sowing.

Variation Sources	DF	Mean Squares					
		AGR_PH_	RGR_PH_	AGR_SD_	RGR_SD_	AGR_LA_	RGR_LA_
Salinity levels (SL)	4	0.323 **	2.1 × 10^−4^ **	0.011 **	8 × 10^−6 ns^	430.53 **	4.4 × 10^−4^ **
Linear regression	1	1.19 **	6.0 × 10^−4^ **	0.039 **	6 × 10^−6 ns^	1500.1 **	1.1 × 10^−3^ **
Quadratic regression	1	0.084 **	2.3 × 10^−4^ **	0.017 **	0 × 10^−5 ns^	54.72 **	8 × 10^−5^ *
Hydrogen peroxide (H_2_O_2_)	4	1.5 × 10^−3 ns^	5.0 × 10^−6 ns^	4.0 × 10^−4^ *	3 × 10^−6 ns^	22.15 **	8 × 10^−5^ **
Linear regression	1	5.2 × 10^−3 ns^	1.7 × 10^−5 ns^	1.7 × 10^−4 ns^	6 × 10^−6 ns^	4.40 ^ns^	0 × 10^−5 ns^
Quadratic regression	1	2.0 × 10^−4 ns^	4.0 × 10^−6 ns^	9 × 10^−5 ns^	0 × 10^−6 ns^	0.038 ^ns^	9 × 10^−5^ *
Interaction (SL × H_2_O_2_)	16	6.4 × 10^−3^ **	7.0 × 10^−6 ns^	2.4 × 10^−4 ns^	3 × 10^−6 ns^	17.32 **	3.9 × 10^−5^ *
Blocks	2	1.8 × 10^−3 ns^	2.1 × 10^−5 ns^	1.3 × 10^−3^ **	5 × 10^−6 ns^	6.18 ^ns^	2.0 × 10^−5 ns^
Residual	74	2.1 × 10^−3^	6.0 × 10^−6^	1.5 × 10^−4^	4 × 10^−6^	2.27	2.0 × 10^−5^
CV (%)		11.0	20.0	16.0	20.1	15.9	24.5

** significant at *p* ≤ 0.01; * significant at *p* ≤ 0.05; ^ns^ not significant.

**Table 3 plants-13-00390-t003:** Summary of the analysis of variance for root dry mass (RDM), stem dry mass (SDM), leaf dry mass (LDM), and total dry mass (TDM) of cotton plants subjected to irrigation water salinity and hydrogen peroxide concentrations at 130 days after sowing.

Variation Sources	DF	Mean Squares			
		RDM	SDM	LDM	TDM
Salinity levels (SL)	4	143.38 **	719.95 **	179.04 **	2648.41 **
Linear regression	1	522.59 **	2597.33 **	584.54 **	9998.61 **
Quadratic regression	1	49.23 **	168.69 **	2.96 ^ns^	334.50 **
Hydrogen peroxide (H_2_O_2_)	4	1.94 **	29.45 **	5.94 **	57.99 **
Linear regression	1	2.50 *	0.47 ^ns^	0.019 ^ns^	5.80 ^ns^
Quadratic regression	1	2.46 *	15.01 **	9.87 ^ns^	5.31 ^ns^
Interaction (SL × H_2_O_2_)	16	2.90 **	13.01 **	5.08 **	29.89 **
Blocks	2	0.16 ^ns^	2.09 ^ns^	0.16 ^ns^	4.23 ^ns^
Residual	74	0.48	1.07	1.17	3.61
CV (%)		12.1	7.79	13.7	7.06

** significant at *p* ≤ 0.01; * significant at *p* ≤ 0.05; ^ns^ not significant.

**Table 4 plants-13-00390-t004:** Summary of the analysis of variance for the number of bolls (NB), average boll weight (ABW), seed cotton weight (SCW), lint cotton weight (LCW), and fiber yield (RF) of cotton subjected to irrigation water salinity and hydrogen peroxide concentrations, at 130 days after sowing.

Variation Sources	DF		Mean Squares			
		NB	ABW	SCW	LCW	RF
Salinity levels (SL)	4	445.53 **	2.33 **	2128.83 **	577.45 **	367.93 **
Linear regression	1	1706.30 **	3.07 **	7935.93 **	2082.08 **	141.67 ^ns^
Quadratic regression	1	35.21 **	1.03 **	191.71 **	55.94 **	1110.62 **
Hydrogen peroxide (H_2_O_2_)	4	26.25 **	1.03 **	71.56 **	19.74 **	258.16 *
Linear regression	1	34.56 **	0.0094 ^ns^	1.22 ^ns^	3.82 *	275.07 *
Quadratic regression	1	1.24 ^ns^	3.25 **	55.56 **	29.41 **	105.59 ^ns^
Interaction (SL × H_2_O_2_)	16	14.60 **	0.46 **	92.17 **	11.99 **	280.39 **
Blocks	2	3.0 ^ns^	0.56 ^ns^	0.30 ^ns^	0.45 ^ns^	116.77 ^ns^
Residual	74	1.16	0.13	2.79	0.63	70.65
CV (%)		14.5	19.3	11.0	10.2	16.6

** significant at *p* ≤ 0.01; * significant at *p* ≤ 0.05; ^ns^ not significant.

## Data Availability

Data are contained within the article.

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
