# Peer review of "Hydrogen Peroxide Alleviates Salt Stress Effects on Gas Exchange, Growth, and Production of Naturally Colored Cotton"

_plants, 2024, doi:10.3390/plants13030390_

Round 1

Reviewer 1 Report

Comments and Suggestions for Authors

The manuscript Hydrogen peroxide alleviates salt stress effects on gas exchange, growth and production of naturally colored cotton submitted by Nóbrega et al. had analysed the effects of hydrogen peroxide on naturally colored cotton to improve its tolerance to salt stress. They used two factor randomized block experiments, and found that under conditions of low water salinity (0.3 dS m-1), foliar application of hydrogen peroxide stimulated the biomass formation and production components of cotton. This manuscript holds certain significant guidance for irrigation with salt water in the cultivation of naturally colored cotton.

There are two concerns in this manuscript:

1, Please indicate their full names where abbreviations first appear. There are many abbreviations in this manuscript such as “ECw”, “DAS” and so on.

2, I think that the authors should cite more literature in the introduction and discussion, and deeply explore the role of H2O2 in plants tolerance to stress.

Author Response

 Dear Editor
The authors are very grateful to you and the Reviewers for the positive and constructive
comments and suggestions on our manuscript entitled “Hydrogen peroxide alleviates salt
stress effects on gas exchange, growth and production of naturally colored cotton”. The
authors would like to inform you that a thorough revision of the manuscript was made,
incorporating the suggestions and adapting the text according to the comments. Attached
is the revised version of the manuscript. All changes are highlighted in red.
The authors would like to express their great appreciation for the comments on our article
and remain at your disposal for any further information and explanation.
The responses/clarifications to the issues raised by the Reviewers/Editor are presented
below:
REVIEWER 1
Commentary 1:
Please indicate their full names where abbreviations first appear. There
are many abbreviations in this manuscript such as “ECw”, “
DAS” and so on.
Response: The suggestion was accepted in the revised version of the manuscript, “salinity
levels of irrigation water – ECw” and “
days after sowing (DAS)”.
Commentary 2: I think that the authors should cite more literature in the introduction
and discussion, and deeply explore the role of H
2O2 in plants tolerance to stress.
Response: As suggested, the following paragraphs were added to the introduction:
Thus, pre-treatment or exposure, such as foliar application with H
2O2 in plants, can induce
or modulate cross-tolerance, through the physiological processes involved, such as in the

photosynthetic process, as well as in various pathways responsive to stress conditions,
such as in ROS and antioxidant metabolism, reducing oxidative stress and lipid
peroxidation [14].
Endogenous application of H
2O2 is associated with overexpression of antioxidative
enzymes and increased levels of ascorbate and gutathione, associated with the
accumulation of osmoprotective substances with proline and soluble carbohydrates,
improving plant tolerance to stress conditions [15].
As suggested, the following paragraphs were added to the discussion:
Furthermore, this stomatal regulation mechanism leads to a reduction in the flow of
water vapor, aiming to maintain water potential and prevent dehydration of guard cells,
which ends up inducing a lower influx of CO2 into leaf mesophyll cells, reducing
transpiration [29].
At low levels, hydrogen peroxide has the ability to quickly diffuse into the
membrane, reaching the coordination of several signaling molecules, increasing the
accumulation of osmoprotective substances and the plant's antioxidative defense system
[32, 33], as observed in our research, where gas exchange was better close to 50 µM.

Reviewer 2 Report

Comments and Suggestions for Authors

Dear author,

The manuscript can be improved with the following minor comments;

First, the introduction section properly closes with an explicit goal. I just would recommend adding a short sentence before to emphasize the research gap that inspired pursuing this work. Please read it to improve the introduction (https://doi.org/10.1080/15440478.2023.2282048).

Please write in detail how you maintained the salt stress throughout the experiment. Please read the above-mentioned article carefully.

In the method section please add some more recent references.

Please also add details about the genotype used in this study is salt tolerant, susceptible, or commercial line?

Since abiotic stresses are usually pleiotropic, please also comment on biomass allocation and reproduction output (yield) in the face of drought, which is a stress typically correlated with salt in the face of changing climate (please refer to and include (https://doi.org/10.3389/fpls.2023.1265700). Also related to this point on pleiotropy in abiotic stress responses, authors must briefly comment on the expected biomass/reproductive trade-off, evident in subtle ways such as plant architecture and seed nutrient accumulation (https://doi.org/10.3390/agronomy12061310). 

Please envision any other recommendation by adding a short perspectives section before the conclusions.

Comments on the Quality of English Language

Minor improvement is required

Author Response

Dear Editor

The authors are very grateful to you and the Reviewers for the positive and constructive comments and suggestions on our manuscript entitled “Hydrogen peroxide alleviates salt stress effects on gas exchange, growth and production of naturally colored cotton”. The authors would like to inform you that a thorough revision of the manuscript was made, incorporating the suggestions and adapting the text according to the comments. Attached is the revised version of the manuscript. All changes are highlighted in red.

The authors would like to express their great appreciation for the comments on our article and remain at your disposal for any further information and explanation.

The responses/clarifications to the issues raised by the Reviewers/Editor are presented below:

REVIEWER 2

Commentary 1: First, the introduction section properly closes with an explicit goal. I just would recommend adding a short sentence before to emphasize the research gap that inspired pursuing this work.

Response:  Following what was suggested, the following paragraph was added to the introduction:

Despite being a culture that has great importance for Semi-arid regions, such as the Brazilian Northeast, little information is available on strategies such as foliar application of H2O2 to improve the performance of fiber cotton plants naturally under saline stress conditions, which motivated the development of this research.

Commentary 2: Please write in detail how you maintained the salt stress throughout the experiment. Please read the above-mentioned article carefully.

Response: Following the evaluator's comment, the following excerpt was added to the material and methods:

To ensure that the plants were under the indicated salinity conditions, the electrical conductivity of the soil was measured every 15 days through the drainage water with the aid of a bench conductivity meter.

Commentary 3: In the method section please add some more recent references.

Response: Farias, F.J.C.; Morello, C. de L.; Pedrosa, M.B.; Suassuna, N.D.; Silva Filho, J.L da; Carvalho, L.P de; Ribeiro, J.L.; BRS JADE: nova cultivar de algodão colorido de dupla aptidão para o cerrado baiano e para o semiárido nordestino. Congresso Brasileiro de algodão. 2017.

Commentary 4: Please also add details about the genotype used in this study is salt tolerant, susceptible, or commercial line?

Response: Following the evaluator's comment, the following excerpt was added to the material and methods:

The naturally colored cotton genotype ‘BRS Jade’ it is considered a moderately salinity tolerant cultivar. from the National Cotton Research Center (CNPA) of Embrapa Cotton was used in the study. The cultivar has light brown fiber, adapted to Cerrado and Semi-Arid conditions, presenting productivity and good fiber characteristics, such as length, uniformity, resistance and micronaire index. Furthermore, it is resistant to some of the main cotton diseases, such as angular leaf spot (Xanthomonas citri subsp. malvacearum) and moderately resistant to common mosaic virus (Abutilon mosaic virus – AbMV) [47].

Commentary 5: Since abiotic stresses are usually pleiotropic, please also comment on biomass allocation and reproduction output (yield) in the face of drought, which is a stress typically correlated with salt in the face of changing climate (please refer to and include (https://doi.org/10.3389/fpls.2023.1265700). Also related to this point on pleiotropy in abiotic stress responses, authors must briefly comment on the expected biomass/reproductive trade-off, evident in subtle ways such as plant architecture and seed nutrient accumulation (https://doi.org/10.3390/agronomy12061310). 

Response: Following the evaluator's comment, the following excerpt was added to the discussion:

Thus, as it affects these cell expansion processes, salt stress affects the architecture of cotton plants, reducing growth rates as well as the accumulation of biomass in plant organs.

Reviewer 3 Report

Comments and Suggestions for Authors

Water shortage and salinity are causing most of the losses for all field crops. Farmers will see a reduced growth rate and a reduced biomass production. Reduced gas exchange rates and reduced water use efficiency are earlier physiological parameters the authors have focussed on.

At low concentrations H2O2 is a messenger regulating plant stress response. But, under stress H2O2 will reach toxic concentrations. In fact, H2O2 is causing most of the damage observed under stress. The beneficial concentration range is extremely narrow. It is influenced by a number of factors, such as light intensity. None of these factors has been measured by the authors. Moreover, the authors have not measured at cellular level any of the physiological and biochemical parameters resulting an inhibition of plant growth. Therefore, the information provided is not complete and does not allow transfer of the results to be used in other projects.

In summary, the manuscript describes a well structured case study but does not provide any new information of a broader relevance. Authors should be encouraged to select a different journal for publication.

Comments on the Quality of English Language

The quality of English language is fine, there are only a few typing errors.

Author Response

Dear Editor

The authors are very grateful to you and the Reviewers for the positive and constructive comments and suggestions on our manuscript entitled “Hydrogen peroxide alleviates salt stress effects on gas exchange, growth and production of naturally colored cotton”. The authors would like to inform you that a thorough revision of the manuscript was made, incorporating the suggestions and adapting the text according to the comments. Attached is the revised version of the manuscript. All changes are highlighted in red.

The authors would like to express their great appreciation for the comments on our article and remain at your disposal for any further information and explanation.

The responses/clarifications to the issues raised by the Reviewers/Editor are presented below:

REVIEWER 3

Commentary: Water shortage and salinity are causing most of the losses for all field crops. Farmers will see a reduced growth rate and a reduced biomass production. Reduced gas exchange rates and reduced water use efficiency are earlier physiological parameters the authors have focussed on.

At low concentrations H2O2 is a messenger regulating plant stress response. But, under stress H2O2 will reach toxic concentrations. In fact, H2O2 is causing most of the damage observed under stress. The beneficial concentration range is extremely narrow. It is influenced by a number of factors, such as light intensity. None of these factors has been measured by the authors. Moreover, the authors have not measured at cellular level any of the physiological and biochemical parameters resulting an inhibition of plant growth. Therefore, the information provided is not complete and does not allow transfer of the results to be used in other projects.

In summary, the manuscript describes a well structured case study but does not provide any new information of a broader relevance. Authors should be encouraged to select a different journal for publication.

Response:  Em relação ao comentário do avaliador, os autores resolveram apontar alguns pontos que mostram a importância do referido artigo.

Cotton cultivation has great relevance for small, medium and large producers in Brazil, being fundamental for the Northeast region, which holds 22% of national production, serving as a source of employment and income generation for the population and with great relevance for the National Gross Domestic Product (GDP).

The edaphoclimatic conditions of the region in many cases are a challenge for producers in the region, where water scarcity is common due to low rainfall and high rates of evapotranspiration, inducing farmers to use irrigation, and in many cases, they are water with high saltis used, which can compromise the development and productivity of the crop.

Therefore, there is a need to develop research like this, aiming to develop viable strategies from an application and economic point of view, in order to provide an alternative for producers to reduce the harmful effects of saline stress. Hydrogen peroxide (H2O2) is an easily accessible and low-cost substance, which when applied correctly helps to mitigate the damage caused by salinity.

Given this context, and the lack of information about its applicability in cotton cultivation, we understand that this research has great relevance for the development of cotton farming in the Brazilian semi-arid region, serving as a source of information for farmers and researchers working in the area, aiming to tackling one of the main problems encountered by producers in the Northeast of Brazil.

Round 2

Reviewer 3 Report

Comments and Suggestions for Authors

All plants suffer from stress when exposed to drought or increased salt concentrations. Stress requires an adjustment of physiological and biochemical parameters so that a new balance is found and the plant can continue to grow. The necessary adaptation mechanisms can react at different speeds and are differently developed depending on plant species. Therefore, plants differ in their stress tolerance and prefer different environments.
As a response to stress, farmers observe reduced growth and lower crop yields. In field tests, stress effects on gas exchange or the increase in biomass can be measured during the growth phase. When the rate of photosynthesis is reduced, an increased formation of reactive oxygen species (ROS) is observed in all plants, because with increasing stress the probability increases that electrons are transferred from excited chlorophyll to O2 and are not used for photosynthesis. ROS formed can be detoxified by antioxidants. However, above a threshold concentration, damage predominates and can even lead to the death of organs and entire plants.
At low concentrations, ROS have an important control function. They regulate gene expression and this way are controling mechanisms for stress adaptation. This gave various research teams the idea of ​​treating plants with ROS (e.g. H2O2) to prepare them for growth under stress. Genes that control stress adaptation were activated before a stressful situation occurred. In fact, the expected positive effects could be experimentally proven. However, two problems arose for practical use in agriculture: (1) Positive effects of treatment with H2O2 only occur in a narrow concentration range. If the concentrations are too high, the membrane, genetic material, and pigment damaging effects predominate. (2) H2O2 solutions are unstable. Their concentration must be checked before use. Corresponding equipment and experience are often not available, especially in small, isolated agricultural businesses. An additional difficulty can arise, especially in arid areas, because plants are already growing under stress and there is already an increased internal level of H2O2.
In summary, the problem for all studies on the use of H2O2 is that experimental conditions must be described very precisely in order to allow the results to be reproduced and used in practice. In addition, the H2O2 concentration used should not be viewed in isolation, but rather its effect must be seen in connection with other physiological and biochemical parameters. This is the only way allowing to adequately interpret the observed H2O2 effect on the physiological balance of plants.
The authors fulfilled these conditions by revising their manuscript. The problems described are adequately addressed. The manuscript is now ready for publication.